# Nanostructures in Hydrogen Peroxide Sensing

**DOI:** 10.3390/s21062204

**Published:** 2021-03-21

**Authors:** Ricardo Matias Trujillo, Daniela Estefanía Barraza, Martin Lucas Zamora, Anna Cattani-Scholz, Rossana Elena Madrid

**Affiliations:** 1Laboratorio de Medios e Interfases (LAMEIN), DBI, FACET, Universidad Nacional de Tucumán, Av. Independencia 1800, 4000 Tucumán, Argentina; trujillo.69@gmail.com (R.M.T.); barrazadaniela86@gmail.com (D.E.B.); mzamora@herrera.unt.edu.ar (M.L.Z.); 2Instituto Superior de Investigaciones Biológicas (INSIBIO), CONICET, Chacabuco 461, 4000 Tucumán, Argentina; 3Walter Schottky Institute and Physics Department, Technical University of Munich, Am Coulombwall 4, 85748 Garching, Germany

**Keywords:** hydrogen peroxide, sensors, biosensors, nanostructures, enzymes

## Abstract

In recent years, several devices have been developed for the direct measurement of hydrogen peroxide (H${}_{2}$O${}_{2}$), a key compound in biological processes and an important chemical reagent in industrial applications. Classical enzymatic biosensors for H${}_{2}$O${}_{2}$ have been recently outclassed by electrochemical sensors that take advantage of material properties in the nano range. Electrodes with metal nanoparticles (NPs) such as Pt, Au, Pd and Ag have been widely used, often in combination with organic and inorganic molecules to improve the sensing capabilities. In this review, we present an overview of nanomaterials, molecules, polymers, and transduction methods used in the optimization of electrochemical sensors for H${}_{2}$O${}_{2}$ sensing. The different devices are compared on the basis of the sensitivity values, the limit of detection (LOD) and the linear range of application reported in the literature. The review aims to provide an overview of the advantages associated with different nanostructures to assess which one best suits a target application.

## 1. Introduction

The development of fast and reliable detection methods for H${}_{2}$O${}_{2}$ is critical in several areas. Titrimetry [1], spectrometry [2], chemiluminescence [3], fluorimetry [4], chromatography [5], and electrochemistry [6] can be applied to detect H${}_{2}$O${}_{2}$. Electrochemical techniques are preferable because of their simplicity, low cost, high sensitivity, and selectivity. One special class among the electrochemical sensors are the enzymatic biosensors, where the electrocatalysis of H${}_{2}$O${}_{2}$ reduction is performed by immobilized enzymes such us Horseradish Peroxidase (HRP) and Hemoglobin (Hb) [7]. However, this class of electrodes show some disadvantages, caused principally by the degradation overtime of the enzymes immobilized on the surface. Therefore, there is strong scientific interest to develop enzymeless sensors using nanostructured materials. In recent years, non-enzymatic electrodes modified with metallic nanoparticles (NP) or nanowires (NWs) such as Pt NPs, Au NPs, Pd NPs and Ag NPs have been widely applied in H${}_{2}$O${}_{2}$ sensing. Figure 1 shows as an example an electrochemical sensor based on palladium nanowires [8]. This class of sensors usually show large specific surface areas, excellent conductivities, and outstanding electrocatalytic activities [9]. Sensors and biosensors incorporating nanomaterials have demonstrated superior performance compared to their conventional counterparts [10]. Therefore, there are more and more attempts to further tailor nanomaterials with organic or inorganic molecules for improving the sensing capabilities [11].

The aim of this review is to gain an overview of the nanomaterials, molecules, polymers, electrodes, and transducing methods used in H${}_{2}$O${}_{2}$ electrochemical sensing devices. In order to compare the performance of the different sensors reported in the literature, the limit of detection (LOD in μM), the linear range of application (LR), and the sensitivity values (in μA·mM${}^{-1}$·cm${}^{-2}$) are listed in different tables throughout the review along with a brief description of the nanomaterials used. In this work, the LOD is defined as the lowest analyte concentration to be reliably detected, i.e., distinguished from blank, as defined by Armbruster et al. [12].

The discussion of the different sensors is organized in each chapter according to the nature of the nanomaterials used. The last chapters deal with devices based on the use of more complex mixed materials, which take advantage of the conducting properties of polymers, graphene, and carbon nanotubes. We report as well on recent results, which focus on the application of nanomaterials to improve the performance of standard bioreceptors for H${}_{2}$O${}_{2}$, based on hemeproteins and HRP.

## 2. Materials Used for Electrocatalytic Hydrogen Peroxide Sensing

### 2.1. Metal Hexacyanoferrates

Prussian blue (PB) is a transition metal hexacyanometalates (Fe${}_{4}^{III}$[Fe${}^{II}$(CN)6]${}_{3}$). Its reduced form, Prussian White (Fe${}_{4}^{II}$[Fe${}^{II}$(CN)6]${}_{3}$), can catalyze hydrogen peroxide at voltages close to 0 V. This feature is of great importance in the sensor field, since it is possible to avoid signals from interference species in real samples, such as glucose or ascorbic acid [13]. The main feature of PB is its structure (Figure 2), where H${}_{2}$O${}_{2}$ can penetrate the crystalline lattice, but bigger molecules cannot. Because of the high catalytic activity of PB towards H${}_{2}$O${}_{2}$, PB is often mentioned in the literature as an “artificial peroxidase”. PB has another interesting feature; it changes colors when its oxidation state varies [14]. This could enable different sensor’s transduction mechanisms. Up to this date, PB-modified electrodes present some difficulties such as poor long-term stabilities and sensor performance dependence on the pH [15].

It is known that PB can be electrochemically deposited on the electrode surface, producing a dense redox active layer [16]. Two other methods for developing PB are worth mentioning, self-assembly [17] and hydrothermal [18]. Both of them are based on wet chemistry. The self-assembly methods offer control over the formation of the PB, whereas the hydrothermal methods are simple and can be applied to obtain PB crystals in high yields [19].

Strictly, it is its reduced form, Prussian White (PW), that has the ability to catalyze the reduction of H${}_{2}$O${}_{2}$ according to Equation (1) [20]:(1)$$K_{4}Fe_{4}^{II}{\left[Fe^{II}{\left(CN\right)}_{6}\right]}_{3(PW)}+2H_{2}O_{2}+4H^{+}=Fe_{4}^{III}{\left[Fe^{II}{\left(CN\right)}_{6}\right]}_{3(PB)}+4H_{2}+4K^{+}$$

The main advantage of electrodeposited PB relies on the fact that hydrogen peroxide can be detected selectively through electrocatalytic reduction in the presence of molecular oxygen, at a low electrode potential, where the influence of the so-called reductants (ascorbate, urate, acetaminophen) on the electrochemical response can be largely avoided [21]. For example, in the work of Sheng [14], monitoring of H${}_{2}$O${}_{2}$ was achivied in a linear range between 4 μM to 1064 μM and with a detection limit of 0.226 μM and with signal-to-noise ratio of 3 (S/N = 3). Interference from glucose, ascorbic acid, dopamine and uric acid was effectively avoided. In this work, they accomplished this by the formation of PB nanoparticles (PBNPs) in polyaniline (PANI) coated halloysite nanotubes (HNTs), from an oxygen-free solution containing 1 mM of FeCl${}_{3}$, 1 mM of K${}_{3}Fe$(CN)${}_{6}$, 0.025 M of HCl, and 0.1 M of KCl used as the supporting electrolyte.

The main drawback of PB-based sensor is its limited stability, especially at neutral pH solutions. When measuring the sensitivity three consecutive times in a pH 7.3 solution, by a drop up to 40% on the third calibration curve with respect to the first one observed. In more acidic solutions (pH 5.2), the stability seems to be better, with a decrease of sensitivity only of 15%. This was observed by Garjonyte et al. under the same experimental conditions [22]. Attempts have been made to optimize the procedure of PB electrodeposition in order to increase the stability [23]. However, since the instability is caused by the chemical degradation of a PB layer during electrocatalytic reduction of hydrogen peroxide, it seems that it cannot be fully avoided [24].

Highest sensitivities are reported for the use of different hexacyanoferrates in combination with other nanostructures. However, this approach results in very good detection limits up to 33 nM but at the cost of losing sensitivity [25,26].

Li et al. [27] developed a H${}_{2}$O${}_{2}$ sensor through the electrodeposition of a Prussian Blue film over a glassy carbon electrode (GCE). What is remarkable here is the sensor’s capability of detecting not only H${}_{2}$O${}_{2}$ but also dopamine. For dopamine, the linear range was: 0.5 μM to 0.7 mM and the limit of detection was 125 nM, while, for H${}_{2}$O${}_{2}$, the linear range was 0.8–500 μM and the limit of detection was 250 nM.

Lin et al. [20] developed a H${}_{2}$O${}_{2}$ sensor based on PB electrodeposited on polypyrrole nanowires (PPy/PB NWs) to obtain a 3D sensor configuration with improved sensitivities with respect to the 2D preparation method. Their analysis clearly showed the increase in sensibility of the PPy/PB NWs sensor. Following a similar approach, Ni et al. [28] employed gold nanoparticles on glassy carbon electrodes as a substrate for electrodepositing PB. This substrate presented two main advantages firstly, gold is a well-known catalyst for the reduction of H${}_{2}$O${}_{2}$ and secondly, the 3D electrode configuration can improve facilitating the contact of H${}_{2}$O${}_{2}$ with the redox centers, therefore improving the sensing performance.

Recently, inkjet printing has became one of the most employed fabrication techniques for sensors optimization. Cinti et al. develped a sensor based on screen printed electrodes (SPEs) modified with PBNPs deposited using piezoelectric inkjet printing [29] The authors electrodeposited different layers of PBNPs in cycles of 4, 5, 10, 20, and over 30 depositions. The best characteristic of Cinti’s fabrication process is the good relation between performance and low-cost production. Figure 3 shows the results obtained with an PBNPs electrodeposition layer of 20 cycles.

Ionic liquid (IL), consisting of anions and cations, have high conductivity and good chemical stability, among other features, at room temperature [30,31]. These properties made IL an attractive material as a modification agent of electrodes for sensing applications. Zhu et al. reported an enzyme free sensor fabricated by doping IL into Prussian blue-multiwalled carbon nanotubes (PB-MWCNTs) [32]. An interesting point to remark is the functionalization method. The authors developed a simple method to fabricate the sensor, and it was characterized by FTIR analysis. The system presented a linear response in the 5–1645 μM range, a sensitivity of 0.436 μA·mM${}^{-1}$·cm${}^{-2}$ and a detection limit of 0.35 μM (S/N = 3). The sensor selectivity was tested with good results in milk samples.

#### Other Metal Hexacyanoferrates

Other transition metal hexacyanate complexes have been successfully applied for H${}_{2}$O${}_{2}$ detection, for example, copper ferricyanide (Cu${}_{2}$[Fe(CN)${}_{6}$]) or potassium nickel hexacyanoferrate (KNi[Fe(CN)${}_{6}$]). Zhang developed a photoelectrochemical sensor for hydrogen peroxide based on nickel(II)-potassium hexacyanoferrate (NiHCF)-graphene hybrid materials. These nanostructures show higher stabilities in slightly basic pH compared to iron hexacyanoferrate. Tria et al. [33] employed the constant potential method to develop a multi-layered structure of PB followed by nickel (II) hexacyanoferrate to stabilize the redox mediator layer in neutral to slightly basic pH solutions. This stability at basic pH was studied by Lin et al. [34] by using a screen-printed electrode. Such electrode was electrochemically modified by electrodepositing nickel hexacyanoferrate (III) (NiHCF) onto the electrode surface using cyclic voltammetry (CV). The NiHCF film has been proven to remain stable after CV scanning from 0 to +1.0 V vs. Ag/AgCl in the pH range of 3 to 10. An interesting study by Zhang et.al. [35] shows that detection of H${}_{2}$O${}_{2}$ can be performed on this class of NiHCF sensors by the quantification of the changes in the photocurrent.

Chromium can be employed to synthesize another type of hexacyanoferrate. Zhang et al. [36] fabricated a Glassy Carbon Electrode (GCE) modified with chromium hexacyanof errate/single-walled carbon nanotube to sense H${}_{2}$O${}_{2}$. This sensor had a linear range from 0.5 μM to 10 mM.

Another transition metal employed in hexacyanoferrates is cobalt. This metal has the advantage of having, as iron, two oxidation states. This generates several combinations of redox states within the same compound. Cobalt hexacyanoferrate was synthetized in the form of nanoparticles by Yang et al. [37]. By depositing the NPs onto a graphene support, the authors developed a nonenzymatic H${}_{2}$O${}_{2}$ sensor. This sensor had a linear range between 0.6 to 379.5 μM and a limit of detection of 0.1 μM. A different fabrication approach was taken by Han et al. [38]. Here, a mixture of cobalt hexacyanoferrate nanoparticles (CoNP) and platinum nanoparticles (Pt) on carbon nanotubes (CNTs) was applied to improve hydrogen peroxide sensing. The sensor showed a linear response of 0.2 to 1.25 mM, a sensitivity of 0.744 A M${}^{-1}$ and response time of 2 s. Other transition metals such as copper [39,40] and ruthenium [26] can be employed to optimize hexacyanoferrates-based materials.

Table 1 shows the performance of sensors that uses metal hexacyanoferrates to detect H${}_{2}$O${}_{2}$. The highest sensitivities in the order of tens of thousands of μA·mM${}^{-1}$·cm${}^{-2}$ were obtained by Han et al. using PB at platinum nanoparticles. This sensibility was a hundred times superior to the PBNPs employed by Cinti et al. or to the CuHCF nanostructures employed by DeMattos et al.

### 2.2. Metallic Nanostructures

H${}_{2}$O${}_{2}$ can be detected electrochemically, either by oxidation or reduction, using metallic catalysts such as platinum (Pt), copper (Cu), and titanium nitride (TiN) [41,42,43,44,45]. Generally, this detection mechanism is based on the application of a certain overpotential (against the reference) to the working electrode, which is around +0.9 V DC when bulk metal electrodes are used [41]. An extremely diverse and interesting alternative for the electrochemical detection of H${}_{2}$O${}_{2}$ are metallic nanomaterials. These materials have among their qualities an excellent conductivity, electro-catalytic activities, higher specific surface and long stability, which offer improved performance as detection interfaces. Such is the case, for example, of gold NPs, which show excellent electrical conductivities and reactivities. On the other hand, the use of polymers as support materials, particularly polydopamine (PDA), allows the successful loading and stabilization of metallic nanoparticles on different substrates, forming films that include organic and inorganic materials.

Hrapovic et al. have reported on the use of metallic nanoparticles for the development of an electrochemical sensor with significantly improved sensitivity for the detection of H${}_{2}$O${}_{2}$. In this work, Pt nanoparticles were used in combination with single-walled carbon nanotubes (SWCNT). The SWCNT were first solubilized in the perfluorosulfonated polymer, Nafion, allowing to significantly reduce the overvoltage for the hydrogen peroxide reaction [46].

Differently, Kang et al. [47], carried out the electrochemical deposition of Au and Pt nanoparticles on a titania nanotubular electrode. The resulting electrodes showed remarkably improved catalytic activities. This class of substrates can be easily prepared and offer very good biocompatibilities. Similarly, the group of Cui et al. [48] took advantage of the ordered anodized titanium nanotubular TiO${}_{2}$ to introduce Pt nanoparticles through an electrodeposition process and obtained a Pt/TiO${}_{2}$ nanostructured electrode for the detection of H${}_{2}$O${}_{2}$.

In addition, using Pt nanoparticles, the group of Chakraborty et al. [49] developed a H${}_{2}$O${}_{2}$ detection platform, using polymers to support metallic nanoparticles. The electrochemical sensing interface has been developed by pre-organizing metal precursors onto a polymer-modified conductive substrate and subsequently chemically reducing the precursor to a nanometric metal. The electrode obtained by this process is very sensitive (9.15 μA/mM) and shows a linear response of up to 4 mM H${}_{2}$O${}_{2}$, and can also detect 0.5 nM (S/N = 5) at an applied potential of 0.5 V.

Gutes et al. [50], on the other hand, used palladium nanoparticles for the amperometric detection of H${}_{2}$O${}_{2}$. This group, by means of an electroless coating of palladium on silicon, obtained palladium films and palladium nanoparticles on silicon substrates. By controlling the immersion time, it was possible to easily synthesize a varied particle density. The substrates are found to exhibit an excellent linear electrocatalytic response down to the micromolar range at 0.0 V, with the lowest quantifiable concentration of 1 μM.

Chen et al. [51] developed a H${}_{2}$O${}_{2}$ sensor with a glassy carbon electrode modified with natural nanostructured attapulgite (ATP) metallic Ag NPs. The resulting sensor reached 95% of steady state current in 2 s and had a H${}_{2}$O${}_{2}$ detection limit of 2.4 μM. This approach uses natural nanostructures that have the advantage of being stable under ambient conditions. This advantage overcomes the high cost of glassy carbon electrodes compared to carbon paste or screen printed electrodes.

Lu et al. [52] developed a H${}_{2}$O${}_{2}$ sensor based on the interactions between DNA strands and a Ag NPs/Graphene Oxide composite. In this particular paper, the reader should pay particular attention to the diverse immobilization methods involved in the fabrication.

Zhang et al. [53] synthesized bimetallic nanoparticles for the amperometric sensor in H${}_{2}$O${}_{2}$ detection. Three bimetallic nanoparticles (platinum + ruthenium, PtRu, platinum + gold, PtAu and platinum + iridium, PtIr) were synthesized by a very interesting method that employs microwave irradiation. Another remarkable feature of these NPs is the fact that two of them, PtRu and PtIr, showed very low response towards a common interfering substance such as ascorbic acid.

Wei-Han Hsiao et al. [9] designed two silver nanostructured sensors for H${}_{2}$O${}_{2}$ based on urchin-like Ag NWs and Ag NPs. Both nanostructured materials were applied on a screen printed carbon (SPC) electrode. The authors developed preparation methods relatively simple for both nanostructures since the reagents were dissolved in aqueous solutions. Sensors based on, urchin-like Ag NWs present better sensitivity with respect to Ag NPs sensors and can be deposited without the aid of compley procedures. Moreover, urchin-like Ag NWs based sensors show better sensitivity and LOD, also at lower applied bias. While urchin-like Ag NWs needed −0.28 V, the Ag NPs needed −0.40 V. This last feature is quite important to minimize the effect of interfering substances and reduce electrode degradation.

Figure 4 shows an SEM image of the urchin-like nanostructures and the calibration curves of the amperometric measurements.

Liu et al. have shown that AgNWs are advantageous over GCE (Glassy Carbon Electrodes) towards the electro-reduction of hydrogen peroxide. Following this approach, the authors achieve an efficient and direct deposition of the nanoparticles prepared in situ with reliable size and morphology control of the nanomaterials obtained. Similarly, Lee et al. used Ag nanoparticles for the electrochemical detection of H${}_{2}$O${}_{2}$ [54]. Through the development of a simple manufacturing method for transparent and flexible Ag nanowire films, this group obtained sensors that showed a reasonable detection limit of 46 μM (S/N = 3) and a fast response time (within 2 s). The sensors presented two linear regions with high sensitivities of 749 μA·mM${}^{-1}$·cm${}^{-2}$ for concentrations from 0.2 to 1.5 mM, and 1640 μA·mM${}^{-1}$·cm${}^{-2}$ for 1.7 to 3.4 mM, both with correlation coefficients higher than 0.99. The sensor presented selectivity for the electro-reduction of H${}_{2}$O${}_{2}$ and stability after prolonged storage. Besides the sensing characteristics, this class of electrodes showed remarkable stability.

In addition to this, in a more recent work, Brzózka et al. [55] showed that the form and amount of silver deposit depend on the cathode material, the presence/absence of the Nafion® layer, and the electrodeposition regime. By electrodeposition of silver particles in a water–glycerin solution containing silver ions and sodium dodecyl sulfate (SDS), on pure metal cathodes and after covering them with a Nafion layer, electrodes with good activity were obtained. Electrocatalytic towards H${}_{2}$O${}_{2}$ reduction compared to a carbon rod electrode Figure 5. The C${}_{rod}$@Ag-Ps electrode showed a high sensitivity of approximately 0.128 μA·mM${}^{-1}$·cm${}^{-2}$ with a detection limit of 0.10 mM and a limit of quantification of 0.33 mM for the detection of H${}_{2}$O${}_{2}$.

Taking the advantage of the great quantity of reactive groups present in proteins, Liua et al. [56] prepared a silver/bovine serum albumin (Ag/BSA) nanocomposite. This nanocomposite was prepared on a core/shell matrix with a fine nanoporous structure, which resulted to be beneficial for mass transfer and heterogeneous catalysis.

Xie et al. [57] reported a one-step method, by Oblique Angle Deposition (OAD), using electron beam evaporation for fabricating TiN nanostructure with tunable morphologies and porosities for electrochemical sensing of H${}_{2}$O${}_{2}$. Amperometric response was investigated by applying −0.2 V. The current had a good linear relationship with the H${}_{2}$O${}_{2}$ concentration in the range of 2.0 × 10${}^{-5}$ to 3.0 × 10${}^{-3}$ M.

Copper has many attractive features in electrode design, but it is prone to oxidization, easily losing its properties. To overcome this problem, Sophia et al. [58] reported the preparation of copper nanoparticles (Cu NPs) stabilized by a polymer, polyvinylpyrrolidone (PVP). By employing a polymer for stabilization, the authors of this work avoided the oxidation of the Cu NPs. Here, the authors achieved a long-term stability, with 85% of the initial sensor response after one month of storage at room temperature. Following the idea of using polymers for loading and stabilizing nanostructures, Li et al. [59] developed an enzyme free sensor based on a gold electrode modified with Au NPs and overoxidized polydopamine composites.

Recently, the group of Cernat et al. [60] has worked on the templateless synthesis of Bi nanowires (BiNW) that involves deep eutectic solvents (DES) as electrolytes, in an attempt to use an ecological approach for the application of Bismuth materials in hydrogen peroxide sensing.

Metal-organic frameworks (MOFs) are very striking materials, whose applicability in sensors is of growing interest. It is composed of metal ions with organic linkers, with interesting characteristics as a high specific surface area, high porosity, topological diversity and high chemical and thermal stability. Hira et al. presented recently an electrochemical sensor based on nitrogen-enriched metal-organic framework (N-CoMOF) for sensitive detection of hydrazine and hydrogen peroxide (TEM image of the N-Co-MOF in the inset of Figure 6 [61]. This material was characterized by different techniques, and the electrochemical behavior evaluated with different concentrations of H${}_{2}$O${}_{2}$. The amperometric study showed a rapid time response of 3 s, with an applied potential of −0.35 V vs. Ag/AgCl in 0.1 M PBS (pH = 7) with the successive addition of 10 μM of H${}_{2}$O${}_{2}$. The developed sensor presented a sensitivity of 0.033 μA·mM${}^{-1}$·cm${}^{-2}$ with a linear range between 15–195 μM. The following Figure shows the chrono-amperometric response of the N-Co-MOF electrode for the successive addition of H_2_O_2_ in 0.1 M PBS (pH = 7) with its corresponding calibration plot.

In Table 2, we summarize a few examples of the use of metallic nanostructures in H${}_{2}$O${}_{2}$ sensing. The sensitivities are shown in decreasing order. Sensors based on Ag NWs ([9,54]) shows higher sensitivity than sensors based on hexacyanoferrates (Table 1) ([29]). The use of these metallic nanostructures also allow for better LODs. Interestingly, by applying Pt NPs Chakraborty et al. [49] could optimize sensors with low sensitivity but a very good LOD.

### 2.3. Metal Oxide Nanostructures

The application of noble metals is restricted by their relatively high prices. In recent years, much cheaper transition metal compound nanomaterials including CuO, CuS, Fe${}_{3}$O${}_{4}$, MnO${}_{2}$, MoS${}_{2}$, NiO, ZnO, and TiO${}_{2}$, have been applied extensively in fabricating highly efficient non-enzymatic H${}_{2}$O${}_{2}$ sensors. Taking into account the scarcity and high costs of noble metals, Kuo et al. [62] focused his work on replacements for such metals. Particularly, he investigated on the use of cobalt manganese oxide in H${}_{2}$O${}_{2}$ sensing.

Metal oxides present good features for sensor development such as thermal stability, irradiation resistance and they are prone to form different nanostructures. Another important property that brings great interest in developing nanosensors with metal oxide materials is their eco-friendly behavior. The main drawback of using these nanostructures is that many of them require an overpotential to be applied on the working electrode to electrocatalytically oxidize or reduce H${}_{2}$O${}_{2}$ for its detection. In particular, manganese dioxide nanomaterials are very interesting for electrochemical applications because they are inexpensive, do not pollute the environment, are abundant in nature, and have the characteristic of having excellent catalytic activity for the decomposition of H${}_{2}$O${}_{2}$. For this reason, they are attractive to be applied in analytical chemistry for the electrochemical detection of H${}_{2}$O${}_{2}$.

It is well known that the structure and surface morphology of nanoparticles give them their special properties, particularly exposed faces, surface charges, etc. Hematite nanoparticles (α-Fe${}_{2}$O${}_{3}$) are quasicubic, and this makes them have a greater activity in the oxidation of CO compared to the floral structure, since mainly the 110 planes are those that are exposed. For its part, thin films of semiconductor metal oxides (MOS) are very useful for the detection of gases and chemical substances such as H${}_{2}$O${}_{2}$ vapors, since their electrical conductivity depends on species such as O${}_{2}$, CO and H${}_{2}$.

Furthermore, in recent years, the use of magnetic nanoparticles in drug delivery and biosensing applications has increased greatly. Hematite (αFe${}_{2}$O${}_{3}$), iron oxide under ambient conditions, is an n-type semiconductor (Eg = 2.1) that has very interesting properties and can be applied to many analytical fields.

A TiO${}_{2}$/SiO${}_{2}$ composite prepared by the sol-gel route can produce highly emissive broadband room temperature phosphorescence at an excitation wavelength of 403 nm. Shu et al. [63] investigated the phosphorescence quenching of a nano TiO${}_{2}$/SiO${}_{2}$ composite as a sensor probe for H${}_{2}$O${}_{2}$. The most interesting about this work is the analysis carried on the nanostructures here developed and the proposed detection mechanism. Such sensor exhibited a linear response to H${}_{2}$O${}_{2}$ concentration ranging from 7.0 × 10${}^{-6}$ to 7.0 × 10${}^{-2}$ M.

Sivalingam et al. [64] fabricated a ZnO thin film sensor by spray pyrolysis technique. These thin films have a nanocrystalline structure where electrical conduction is determined by the electron paths along the grains and the grain boundaries. This property was employed by the authors as a sensing mechanism. Two features are remarkable here, one is the idea of a sensor for H${}_{2}$O${}_{2}$ in a gas state. The second is the study carried on the sensor’s temperature of operation, showing that the sensitivity of such sensor towards H${}_{2}$O${}_{2}$ strongly depends on the operating temperature.

Zhang et al. [65] examined the electrochemical and electrocatalytical behavior of flower like copper oxide modified glass carbon electrodes (CuO/GCE). Interestingly, here we find that the proposed sensor works not only for H${}_{2}$O${}_{2}$ but also for another analyte with biomedical importance, nitrite ion (NO${}_{2}^{-}$). In order to improve the sensing mechanism towards both analytes, especially NO${}_{2}^{-}$, the operating bias potential must be lowered. Employing metal oxides, such as CuO, with electro-catalysis properties is a very good way to achieve this purpose. This shows how versatile some nanostructures can be. In this work, Zhang et al. did not need to use Nafion as a membrane. This is an improvement over two previous works where they employed α-Fe${}_{2}$O${}_{3}$ NPs plus chitosan [66] and Cu${}_{2}$O together with Nafion [67] for the detection of H${}_{2}$O${}_{2}$ and NO${}_{2}^{-}$. Liu et al. [68] also studied α-Fe${}_{2}$O${}_{3}$ as an electrochemical sensor for H${}_{2}$O${}_{2}$, but, in this case, it was synthesized as nanorods.

Li et al. [69] synthesized MnO${}_{2}$/graphene as an electrocatalyst for nonenzymatic H${}_{2}$O${}_{2}$ detection. On the bright side, the presented sensor showed some good features such as fast response. The authors included some ions such as Fe${}^{3+}$, Cu${}^{2+}$, NO${}_{3}^{-}$, etc. The material also presented good long-term stability, losing only 10% of its original response after 30 days. He et al. group [70] also employed MnO${}_{2}$ for H${}_{2}$O${}_{2}$ sensing together with a carbon material as a means to increase electrical conductivity. For that purpose, this group employed carbon foam, which is a 3D self-supported material. What it is interesting about this material is its lower cost and relatively easier fabrication compared to other carbon foams like graphene foam. Employing MnO${}_{2}$ but in a different configuration, Begum et al. [71] synthesized carbon nanotubes with δ-MnO${}_{2}$ (δ-MnO${}_{2}$/CNTs) nanocomposite as enzyme-free sensor for the detection of H${}_{2}$O${}_{2}$ through an electroreduction reaction (Figure 7). Due to the relatively large interlayer distance (ca. 0.7), the δ-MnO${}_{2}$ shows decent electrochemical performances among all polymorphs of MnO${}_{2}$. Among the interesting study carried on this work, one should notice the sensor response at different pHs. In addition, it is noticeable that, besides the normal interference analysis, i.e., adding common interferes to a buffer solution, this group evaluated their sensor’s performance in tap water and in a tomato sauce.

It has been observed that some class of liver cancer cell can rapidly release H${}_{2}$O${}_{2}$ under stimulation. Kong et al. [72] developed a sensor for real-time H${}_{2}$O${}_{2}$ measurements within these cancer cell environment by using Co${}_{3}$O${}_{4}$ NWs over rGO. The oxidation current of H${}_{2}$O${}_{2}$ varied linearly with respect to its concentration from 0.015 to 0.675 mM. The detection limit was 2.4 μM with a sensitivity of 1.14 mA·mM${}^{-1}$·cm${}^{-2}$. Recently, Liu et al. provide a new approach to real-time, in situ, sensitive and rapid monitoring for peroxide detection in a wide linear detection range from 1 to 1000 mM, a high sensitivity of 13.2 μA·mM${}^{-1}$·cm${}^{-2}$, a detection limit of 40.2 μM (S/N = 3) and a fast response time of 5 s. The construction of this new autoamplified bioelectrochemical sensor uses a bioanode as an electrical supplier and a graphite cathode modified with CoMn${}_{2}$O${}_{4}$ nanoparticles (CoMn${}_{2}$O${}_{4}$@GE) as a sensor element for the determination of H${}_{2}$O${}_{2}$. It also has good reproducibility, acceptable selectivity and excellent long-term stability [73].

On the other hand, Lu et al. developed a biosensor with a novel and simple strategy to fabricate a 3D nitrogen doped perforated graphene hydrogel decorated with NiCo${}_{2}$O${}_{4}$ nanoflowers (NHGH/NiCo${}_{2}$O${}_{4}$) through a one-pot hydrothermal method with subsequent calcination. As a result, the proposed hybrid nanocomposite could be a good electrochemical biosensor since it showed wide linear ranges (1–510 μM) and low detection limits (0.136 μM) in alkaline solution (S/N = 3) for the detection of H${}_{2}$O${}_{2}$ [74].

Lee et al. [75] investigated Cobalt oxyhydroxide (CoOOH) nanosheets for electrochemical sensing of H${}_{2}$O${}_{2}$ and hydrazine (N${}_{2}$H${}_{4}$). When synthesizing the CoOOH nanosheets, this group presented a simply wet chemistry approach. Interestingly, the magnetic properties of the materials were used to support stirring in solution to improve sensing. When talking about the sensor’s construction, it is important to remark that this sensor has no need of an entrapment matrix such as Nafion or Chitosan. Furthermore, as mentioned before, it is of great interest to develop sensors with multiple capabilities based on one particular nanostructure. In this paper, the author analyzed the sensor’s characteristics for both analytes independently, with very interesting results. For example, the sensor obtained a reproducibility with a standard deviation lower than 5% for both H${}_{2}$O${}_{2}$ and N${}_{2}$H${}_{4}$.

In a recent work, Mai et al. successfully prepared a highly sensitive and stable H${}_{2}$O${}_{2}$ sensor by direct growth of heterostructure of cobalt oxide-sulfide nanosheets in three-dimensional foam. The structural conformation of this sensor would be key to improve the catalytic performance for the detection of peroxide, resulting in a device with a sensitivity of 0.059 mA·mM${}^{-1}$·cm${}^{-2}$, a wide detection range of 2 to 954 μM, and a limit of low detection of 0.890 μM [76].

Keeping in mind the environmental impact of developing new technologies, Khan et al. [77] investigated an eco-friendly material, Co doped ZnO NPs for its use as an electrochemical sensor for H${}_{2}$O${}_{2}$. The developed sensor displayed a sensitivity of 92.44 μA·mM${}^{-1}$·cm${}^{-2}$ and a limit of detection of 14.3 μM.

Molybdenum disulfide (MoS${}_{2}$) has a sandwich structure of three hexagonal atomic layers (S–Mo–S). Strong covalent bonding among atoms enables the formation of 2D layers, with the layers loosely bound to each other via weak van der Waals interactions [78]. The anisotropic structure of MoS${}_{2}$ leads to significant anisotropy in electrical conductivity. This material is used as lubricant and catalyst for reactions like hydrodesulfurization [79]. This material has the particularity of having a similar structure as graphene and therefore it has been employed in the nanoelectronics fields [80]. One of its best features for electronics is the high on-off ratio, which is comparable to the graphene nanoribbons one. Moreover, the MoS${}_{2}$ structural properties allows ionic intercalation between layers which is desirable for electrochemical sensors because it significantly facilitates electron transfer. Lin et al. [81] constructed a 3D nano-flower-like Cu/multi-layer molybdenum disulfide composite (CuNFs/MoS${}_{2}$) modified glassy carbon electrode to achieve a non-enzymatic sensor for H${}_{2}$O${}_{2}$ (Figure 8). Linear ranges were obtained between 0.04–1.88 μM and 1.88–35.6 μM for H${}_{2}$O${}_{2}$. LOD was 0.021 μM.

Wang et al. achieved an extremely sensitive H${}_{2}$O${}_{2}$ biosensor based on MoS${}_{2}$ nanoparticles with a detection limit as low as 2.5 nM and a wide linear range of 5 orders of magnitude (Figure 9). On the basis of this biosensor, the trace amount of H${}_{2}$O${}_{2}$ released from Raw 264.7 cells was successfully recorded [82].

Recently, Bohlooli et al. presented the great potential of the combination of a transition metal oxide, such as manganese oxide nanomaterial with carbon nanowalls (CNW), to form a composite for an efficient electrochemical sensor for H${}_{2}$O${}_{2}$. The sensor showed very good electrocatalytic activity with a wide linear range between 40–10,230 μM H${}_{2}$O${}_{2}$, high sensitivity (698.6 μA·mM${}^{-1}$·cm${}^{-2}$), and an LOD of 0.55 μM. The only feature that would not be very desirable in this sensor is the relatively high working potential, of +0.7 V vs. Ag/AgCl [83].

Finally, Sankarasubramanian et al. developed a new material for the non-enzymatic electrochemical detection of H${}_{2}$O${}_{2}$, which presented a lower detection limit, a wide linear range, fast response time and high sensitivity towards this analyte. It is important to highlight that the new compound obtained by using a simple chemical spray pyrolysis method of deposition of the semiconductor oxide, CdO (Ti: CdO) doped with Ti, using indium doped tin oxide (ITO) as substrate, shows a pronounced electrocatalytic activity for the reduction of H${}_{2}$O${}_{2}$ compared to pure CdO [84].

The next table (Table 3) summarizes various metal oxide nanostructures used in H${}_{2}$O${}_{2}$ sensing. Co${}_{3}$O${}_{4}$ NWs reported by Kong et al. [72], showed the maximum sensitivity. Although sensors based on these metal oxide nanostructures show great sensitivity, the LODs observed are not as good as the ones presented in Table 1 and Table 2.

### 2.4. Mixed Nanostructures

In this section of the review, we refer to a work in which multiple nanostructures applied on the same device and used as supporting materials for the fabrication of complex nanocomposites and hybrid materials. Complex nanocomposites based on conductive polymers (CP) have electrical and optical properties similar to those of metals or inorganic semiconductors. An interesting example is polypyrrole (PPY), which, in addition to being biocompatible, has high electrochemical activity and conductivity [86].

Other interesting nanocomposites are those based on graphene, due to their known excellent electrical properties. For its part, reduced graphene oxide can be used to manufacture three-dimensional (3D) conductive networks, by using it as a two-dimensional (2D) conductive template over which 1D nanostructures are assembled [87]. Layer-by-layer (LBL) assembly is also used to prepare versatile ultra-thin functional films, very interesting to apply in sensors, separation devices, optics, and electronics [88].

Halloysite nanotubes (HNTs) are attracting interest in material science applications. HNTs are novel and abundant non-toxic natural nanotubes, of high porosity and low cost. They are natural aluminosilicates (Al${}_{2}$Si${}_{2}$O${}_{5}$(OH)${}_{4}$ nH${}_{2}$O) characterized by have nano- and mesopores and can be applied in H${}_{2}$O${}_{2}$ sensing [89].

The use of complex nanocomposites based on the use of transition metal oxides such as Co${}_{3}$O${}_{4}$ offers great advantages in H${}_{2}$O${}_{2}$ sensing. For example, transition metal oxides, such as Co${}_{3}$O${}_{4}$, show excellent electrocatalytic performance toward H${}_{2}$O${}_{2}$ [90]. However, their poor conductivity presents low detection currents compared with noble metals [91]. To overcome this problem, it is possible to use conductive materials like metals or polymers to prepare hybrid compounds.

Nanocomposites using carbon nanotubes have also been widely applied. Single-walled nanotubes (SWNT) have a diameter between 1 and 3 nm with a length of several micrometers [92]. Multi-walled (MWNT) nanotubes are concentric coaxial nanotubes [93]. These nanomaterials have revolutionized materials technology due to their excellent mechanical properties, high electrical conductivity, and large surface area [94]. This last characteristic not only facilitates the transfer of electrons from the nanocomposite but also the mass transport of reactants.

Complex nanocomposites with CNTs are often prepared by using the metal impurities buried in the structure as platform for further functionalization. By following this approach, Gayathri et al. reported the electro-assisted complexation of iron impurity in the MWCNT (MWCNT-Fe) with the amino functional group of chitosan (H${}_{2}$N-CHIT) ([MWCNT-Fe:H${}_{2}$N-CHIT]). The preparation of the sensor is schematically depicted in Figure 10. The modified glassy carbon electrode showed a highly redox-active characteristic, which is similar to that of the redox and functional behaviors of heme peroxidase. This approach was shown to be efficient in H${}_{2}$O${}_{2}$ electrocatalytic and electrochemical sensing applications [95].

Lin et al. [96] developed an enzyme-free H${}_{2}$O${}_{2}$ sensor that consisted of a MWCNT-PEDOT film, which they efficiently deposited on glassy carbon (GCE) and on indium tin oxide (ITO) electrodes. The electrode showed a linear response for different concentrations of H${}_{2}$O${}_{2}$ in the range between 0.1 and 9.8 mM by applying a potential of –0.5V against Ag/AgCl (Figure 11). The electrode had two linear work zones depending on the H${}_{2}$O${}_{2}$ concentration, with a sensitivity of 943 μA·mM${}^{-1}$·cm${}^{-2}$ with a signal/noise ratio (S/N) of 6 for one range, and 174 μA·mM${}^{-1}$·cm${}^{-2}$ with a S/N ratio of 4 for the other. Measurements were made in PBS pH 7. In this particular case, the sensor is also capable of detecting ascorbic acid, dopamine, and uric acid. This capability is of particular interest since the authors manage to discriminate the potential at which each analyte could be detected.

Lin et al. [20] developed a hydrogen peroxide sensor based on a polypyrrole nanowire (PPy)/PB and compared it to PB deposited over a PPy film. The introduction of the PPy nanowire can provide a high surface area and improve the sensitivity to H${}_{2}$O${}_{2}$. PPy fibers are quite interesting mainly because of their stability in ambient conditions and high conductivity.

Polyethyleneimine (PEI) is a water soluble polymer with amine groups. Due to its active amine groups, PEI can react with other materials with some certain active groups, such as carboxyl or epoxy groups. Shan et al. [97] took advantage of these active groups to covalently graft graphene sheets via the nucleophilic ring-opening reaction between the amine groups in PEI and epoxy groups of graphene oxide (Figure 12). The sensor was then constructed by a layer-by-layer complexation method by taking advantage of the difference in electrical charge of the materials. Very much like drop cast, this construction method is simple and easy to be performed. The resulting sensing performance is shown in Figure 13.

Zöpfl et al. [98] studied graphene materials with low number of defects as a signal enhancer for H${}_{2}$O${}_{2}$ detection. Here, a very interesting comparison among three different two-dimensional carbon nanomaterials was carried on with respect to standard measurements on carbon disc electrodes. For this purpose, the graphene materials in this study were prepared by mechanical exfoliation (single layer graphene, SG), chemical vapor deposition (chemical vapor deposited graphene, CVDG), and chemical exfoliation (reduced graphene oxide, rGO), comprised of a different degree of defects. This showed that the quality of the carbon nanostructure has a quantitative impact on the performance of the detection. One of the most interesting conclusions of this study is the fact that the increase in defects on the nanostructure did not result in an improvement of the signal compared to the carbon disc electrode.

One of the best features of the halloysite nanotubes (HNTs) is their natural occurrence. As many other nanostructures, their main disadvantage is the lack of conductivity. Thus, Shen et al. [14] employed a conductive polymer, Polyaniline to modify the HNTs. One of the interesting points of this work is the sensor construction, where HNT modification was carried on in a one-step in situ process, and this nanocomposite was casted onto the electrode, followed by PB electrodeposition. This sensor showed a good performance when determining H${}_{2}$O${}_{2}$ in tap water. It is important to regard that measurements can be carried on at 0V bias, which reduces the interfering effect of other analytes within complex samples. Another example of using natural occurring structures can be found in the work of Gayathri et al. [95]. By following an interesting approach, this group optimized a complex sensing material based on carbon nanotubes, iron impurities, and chitosan in order to mimic the heme-peroxidase function. This biomimetic system followed Michaelis–Menten-type reaction kinetics for the H${}_{2}$O${}_{2}$ reduction reaction with a K${}_{M}$ value of 0.23 mM.

Conducting polymers in nanocomposites have the ability to mediate as well as to decrease the over potential of biomolecular oxidation/redox process. Chan et al. [85] reported the preparation of a nanocomposite film containing zirconia nanoparticles (ZrO${}_{2}$ NPs), poly(toludine blue O) (PTBO), and gelatin functionalized multiwalled carbon nanotubes (GCNT) for amperometric H${}_{2}$O${}_{2}$ sensors. This sensor presented a linear range of 0.05 mM–0.25 mM and a sensitivity of 82.13 μA·mM${}^{-1}$·cm${}^{-2}$.

Table 4 presents some applications of mixed nanostructures in H${}_{2}$O${}_{2}$ sensing. Among the mixed nanostructures showed in this table, sensors based on the use of single graphene (SG) (low-defect graphene) presents the highest sensitivity. Generally, this class of sensors often show high sensitivities, but cannot achieve interesting limit of detections, as shown in Table 4.

### 2.5. Biomolecules

Biosensors that employ enzymes as the biological recognition element can display not only very good sensitivities and selectivities but are often very versatile and can be applied for the detection of different targets. However, this class of biosensors suffer from some main drawbacks linked to the lack of stability and activity of the bioreceptor of choice after immobilization on the solid support [99]. Unfavorable orientation or direct adsorption of biomolecules onto a metal electrode surface may dramatically lower their catalytic activity and the performance of a biosensor mainly depends on the properties of the bioactive layer associated with the transducer. The immobilization step is of paramount importance in biosensing because the biorecognition molecule must retain its functions [100]. There are three generations of biosensors involving electron transfer between the biological element (enzymes) and the bulk electrode. Very briefly, a first generation biosensor is based on the detection of the products of an enzymatic reaction or on natural mediators, i.e., oxygen with the limitation of its concentration dependence. The second generation biosensor is based on redox mediators artificially introduced such as ferrocene. The third generation biosensor is obtained when there is a direct electron transfer between the enzyme and the electrode. The main problems with the second generation biosensors are the potential leaching and toxicity. The last sensing approach is the most difficult to perform and have found only recently concrete applications. Nevertheless, the arrival of the nanotechnology allowed the exponential increase in third generation biosensors research [101].

For third-generation biosensors, the main advantage is the absence of mediators, which provides them with a superior selectivity and sensitivity. In addition, the possibility of modulating the desired properties of an analytical device using protein modification with genetic or chemical engineering techniques.

The mechanism of direct bioelectrocatalytic reduction of H${}_{2}$O${}_{2}$ at HRP-modified electrode is based on the following equations:(2)$$HRP-Fe^{III}+H_{2}O_{2}\to^{K_{1}}CompoundI+H_{2}O$$
(3)$$CompoundI+2H^{+}+2e^{-}\to^{K_{2}}HRP-Fe^{III}+H_{2}O$$

Compound I is an intermediate (oxidation state +5) consisting of oxyferryl iron (Fe${}^{4+}$ = O) and a porphyn π cation radical.

The active site of the HRP enzyme is more or less superficial; therefore, it would be easier to manufacture 3rd generation biosensors using this enzyme. There are several works where the reaction of peroxidases enzymes were revised more thoroughly [102,103]. At high concentrations of H${}_{2}$O${}_{2}$, the reaction results in a higher oxidation state (+6), where the enzyme is inactive [104]. This is a very important parameter to take into account when studying the biosensor. Other electrochemical mechanisms were proposed based in the property of the HRP that can act as reductant or oxidant [105,106].

We have evaluated nearly 100 reports on biosensors for the quantification of H${}_{2}$O${}_{2}$ published in the last 10 years, and we have selected the four most outstanding in terms of sensitivity and LOD in order to make a critical comparison with the enzymeless sensors. We have prepared a table with the most significant related with sensitivity and LOD (see Table 5), but we present here those that we consider important to highlight in order to compare with what interests us in this review.

In particular, for H${}_{2}$O${}_{2}$ sensors, it is also important to state the polarization as well. This is the overpotential applied to the working electrode referred to a reference electrode, e.g., Ag/AgCl, SCE, etc. The higher this potential, the higher the H${}_{2}$O${}_{2}$ signal, but this also increases the possibilities of detecting other electroactive molecules. Furthermore, high overpotentials between 0.5 and 1 V contribute to electrodes degradation.

The combined use of nanostructures (mainly Ag, Au and Pt) with proteins capable of catalyzing H${}_{2}$O${}_{2}$, such as HRP or Hb, can lead to the development of sensors with high sensitivities and LODs. In this context, below, we will carry out the analysis of four interesting works in this area, which use different enzymes and strategies for the biodetection of H${}_{2}$O${}_{2}$ with very good results. Yao et al. [107], successfully fabricated a stable and sensitive biosensor for H${}_{2}$O${}_{2}$ detection, as well as a potential platform for general biosensor design. For the biodetection of this analyte, the HRP enzyme was immobilized between silver nanoparticles (AgNP) and conductive polymers such as poly (3,4-ethylenedioxythiophene): poly (styrenesulfonate) (PEDOT: PSS) to give rise to a sandwich-type biosensor. The biosensor presented an excellent electrocatalytic capacity for H${}_{2}$O${}_{2}$ a good linear relationship of current response in the range of 0.05 to 20 μM and a detection limit of 0.02 μM (S/N = 3). Another example is the biosensor of Yu et al. [108], who suggested that the biological activity of immobilized Hb would significantly improve through the use of core-shell NPs (Fe${}_{3}$O${}_{4}$@Pt). This nanomaterial is an excellent alternative for the biodetection of compounds such as H${}_{2}$O${}_{2}$ and nitrite. This working group particularly developed a third generation biosensor based on LBL films made with Hb and these core–shell NPs (Fe${}_{3}$O${}_{4}$@Pt). This biosensor showed good sensitivity, good reproducibility, a low detection limit (LOD of 0.03 pM) with a wide linear range (0.125 pM to 0.16 mM) and a fast response for the detection of H${}_{2}$O${}_{2}$. In this case, the combination of the assembly of Fe${}_{3}$O${}_{4}$@Pt and LBL for the direct electrochemistry of redox proteins would be the appropriate strategy to improve the sensitivity and stability of the biosensor avoiding the use of mediators. Nandini et al. [109] focused their work on the development of an electrochemical biosensor that uses the enzyme catalase (CAT) and one-dimensional gold nanostructures (1D-AuNs), in a graphite electrode (GE), for the efficient and reliable biodection of H${}_{2}$O${}_{2}$. Compared to previously reported electrochemical biosensors, the proposed biosensor showed higher performance, as it has a fast response (5 s), wide linear range (0.05–19.35 mM), high sensitivity (992 μA·mM${}^{-1}$·cm${}^{-2}$), low detection limit (0.98 nM), and a high enzymatic affinity (Km app = 0.21 mM) for H${}_{2}$O${}_{2}$. Furthermore, it exhibited excellent anti-interference ability, good repeatability, reproducibility, and admirable long-term electrochemical stability. The great performance of the biosensor was achieved thanks to the AuN, which offers a large surface area, biocompatible with CAT, and a multilayer film per layer that facilitates the direct transfer of electrons and, in turn, prevents other molecules from sticking to the electrode, thus maintaining its stability.

Bollella et at. presented an interesting comparison between the use of two different peroxidases enzymes to make a hydrogen peroxide biosensor (Figure 14), the cationic horseradish peroxidase (HRP), and the anionic tobacco peroxidase (TOP), combined with a highly cationic osmium polymer [Os(4,4${}^{\prime }$-dimethyl-2,2${}^{\prime }$-bipyridine) 2poly(N-vinylimidazole)10Cl]${}^{+2/+}$ ([Os(dmp)PVI]${}^{+/2+}$) [110]. They immobilized both enzymes separately onto graphite rod (G) electrodes through a three-step drop-casting procedure, to make two biosensors, one named as HRP/PEGDGE/[Os(dmp)PVI]${}^{+/2+}$/G and the other as TOP/PEGDGE/[Os(dmp)PVI]${}^{+/2+}$/G electrodes, with each enzyme, respectively. The poly(ethyleneglycol) diglycidyl ether (PEGDGE) was used as a cross linking agent between the redox polymer and the enzymes. The electrocatalytic current obtained for TOP enzyme compared to that of HRP was twice higher. The authors indicate that this fact reflects that the electrostatic attraction between the anionic TOP and the cationic redox polymer enhances the electron transfer rate and that the lower degree of glycosylation and molecular weight of this enzyme may facilitate a tighter contact between the active site and the electrode. The TOP biosensor also showed a higher sensitivity (470 × 10${}^{3}$μA·mM${}^{-1}$·cm${}^{-2}$), higher k${}_{cat}$, lower KM${}^{app}$ (302 ± 33 μM), and an improved long-term stability (decrease in 17.3% upon 30 days compared with 50% for HRP). Both peroxidase modified electrodes exhibited a wide dynamic response range (1–500 μM), a low detection limit (0.3 μM), and a fast response of 3 s. The following figure shows the schematic representation of both biosensors, the one that uses the HRP enzyme, and the other that uses the TOP enzyme. The figure also shows the calibration curves for both biosensors.

Finally, in a more recent work, Lee et al. developed a biosensor that allows the determination of hydrogen peroxide in traces, with a detection limit of 100 fM, with a fast response time and high sensitivity. This working group used a highly sensitive single-layer graphene-based field-effect transistor (FET) as a conductive substrate material and cytochrome c (Cyt c) as a biomolecular receptor for H${}_{2}$O${}_{2}$ detection (Figure 15) [111]. Thus, as we can see in Table 5, Lee’s development leads the ranking of the 10 biosensors with the highest sensitivity for the detection of H${}_{2}$O${}_{2}$, analyzed in this review. Furthermore, the development of Yao et al. is also included in this ranking, within the five most sensitive biosensors.

Based on the above, and taking into account developments such as those of Nieto et al. [112], Murphy et al. [113], and Fatima et al. [114], we consider a promising future for the combined use of nanostructures with proteins capable of catalyzing H${}_{2}$O${}_{2}$.

**Table 5 sensors-21-02204-t005:** High sensitivity nanomaterials for peroxide sensing.

Nanomaterial	Transd. Princp.	Sensitivity μA·mM${}^{-1}$·cm${}^{-2}$	LR [mM]	LOD μM	Ref
Biom. (AgNPs)	CV	3 × 10${}^{5}$	0.05–20 × 10${}^{-3}$	0.02	[107]
HRP-Osmium polymer	Amp	470 × 10${}^{3}$	1–500 × 10${}^{-3}$	0.3	[110]
Mixed (SG)	–	202 × 10${}^{3}$	–	651.5	[98]
Metal Hex. (PB@PtNPs/GF)	Amp	40.9 × 10${}^{3}$	–	1.2 × 10${}^{-3}$	[25]
Biom. (Hb/Fe${}_{3}$O${}_{4}$@Pt)	Amp.	12 × 10${}^{3}$	0.125 × 10${}^{-3}$–0.16	0.03	[108]
Metalic (Ag NWs)	Amp	4.705 × 10${}^{3}$	50 × 10${}^{-3}$–10.35	10	[9]
Metalic (nano Pd)	Amp	1.42 × 10${}^{3}$	1–14 × 10${}^{-3}$	1	[50]
Metal Ox. (Co${}_{3}$O${}_{4}$ NWs)	Amp	1.14 × 10${}^{3}$	0.015 to 0.675	2.4	[72]
Metal Hex. (PBNPs)	Amp	0.762 × 10${}^{3}$	0–4.5	0.2	[29]
Metal Ox. (δ-MnO${}_{2}$/CNTs)	Amp	243.9	0.05 to 22	1	[71]
Mixed (CVDG)	–	173	–	15.1	[98]
Cyt c/Graphene FET	Amp	–	100 × 10${}^{-12}$–100 × 10${}^{-9}$	100 × 10${}^{-9}$	[111]

## 3. Discussion

To end this review, we consider that it would be interesting to make a top ten comparison between the analyzed nanostructures that have a high sensitivity in each group. In Table 5, the reader can find a quick view of the similarities and differences between the different nanostructures for peroxide sensing. In this way, we can see that nanomaterials based on biomolecules such as AgNP have the highest sensitivity for peroxide sensing (3 × 10${}^{5}$μA·mM${}^{-1}$·cm${}^{-2}$); however, their nanostructure has the lowest LOD (0.02 [μM]) [107] in this top ten. Interestingly, mixed nanostructures are a good alternative when analyzing their sensitivity values (202 × 10${}^{3}$μA·mM${}^{-1}$·cm${}^{-2}$) and LOD (651.5 [μM]) for peroxide sensing in the case of SG [98]. Another nanostructure that stands out for having good performance characteristics are the metallic ones, such as AgNWs that have a sensitivity of 4.705 × 10${}^{3}$μA·mM${}^{-1}$·cm${}^{-2}$ and an LOD of 10 [μM] [9].

## 4. Conclusions

The presented review provides a general overview of the different nanostructured materials that is used for hydrogen peroxide sensing. Although this is a rapidly expanding field of research, this review clearly shows that the use of different class of nanomaterials can be of great advantage in order to obtain advanced devices possessing improved electrochemical properties in H${}_{2}$O${}_{2}$ sensing. The application of hexacyanoferrate-based nanomaterials is discussed in Section 2.1. Electrodes modified with Prussian Blue (PB), in particular, are the most effective in hydrogen peroxide sensing, thanks to the catalytic properties of this material. Additionally, detection based on PB can be carried out at low electrode potentials, which allows for avoiding the interference of common reductants present in complex biological samples. Interestingly, recent work has shown that the observed low stabilities at neutral pH solutions of PB-modified nanostructures can be overcome by combining regular PB with other transition metals like copper or nickel. In Section 2.2, we discussed another important class of materials for the detection of H${}_{2}$O${}_{2}$, nanostructured noble metals. According to our literature review, noble metals are widely the most applied in the detection of hydrogen peroxide. However, over potentials up to +0.9 V are needed to reduce H${}_{2}$O${}_{2}$ if noble metals are employed in classical detection schemes. Nanostructuring can overcome this problem and detection can be achieved at lower polarization ranges as shown in several contributions discussed in Section 2.2. On the other hand, the scarcity and high cost of noble metals are a major obstacle to the development of detection methods applicable on a large scale. In this sense, detection schemes based on the use of metal oxides can be of great advantage, as discussed in Section 2.3. Metal oxide -based nanomaterials exhibit good characteristics for sensor development because of the high thermal stabilities observed. Devices based on the use of this class of nanomaterials show the highest sensitivities in the literature so far. However, corresponding LODs are significantly lower than expected and cannot compare with the performance of metal hexacyanoferrates. Section 2.4 deals with devices based on the application of complex nanocomposites and hybrid materials. Interesting work has been reported in the literature showing the advantage of exploiting the excellent electrocatalytic features of compounds such as the transition metal oxides in nanocomposites containing polymers such as PEDOT, PEI, or CNTs to ensure for high conductivities. Extremely good sensitivities and detection limits are observed, in particular, for hybrid materials based on the use of graphene. Biosensors discussed in Section 2.5 are based on the use of biomolecules, mostly enzymes with catalytic centers, and generally allow for H${}_{2}$O${}_{2}$ detection with high sensitivities. Selected examples are discussed from the huge number of contributions in the field, focusing on third generation biosensors, which show higher long-term stabilities and activities. In conclusion, this review clearly shows that electrochemical sensors that take advantage of material properties in the nano range can be of high advantage in many applications.

## Figures and Tables

**Figure 1 sensors-21-02204-f001:**
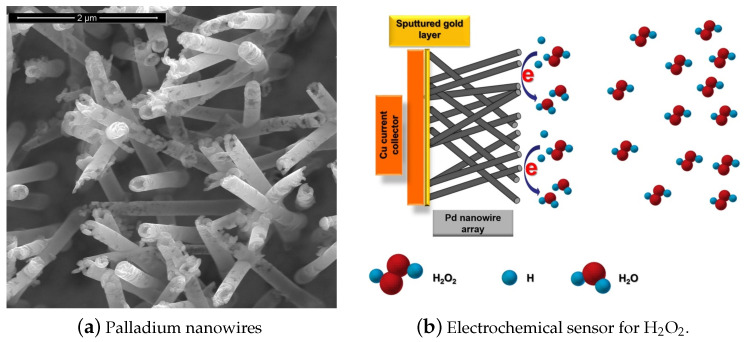
Electrochemical hydrogen peroxide (H${}_{2}$O${}_{2}$) sensor based on palladium nanowires (PdNWs). (**a**) SEM micrograph of the PdNWs electrode, (**b**) basic diagram of the sensor operation. With permission from [8].

**Figure 2 sensors-21-02204-f002:**
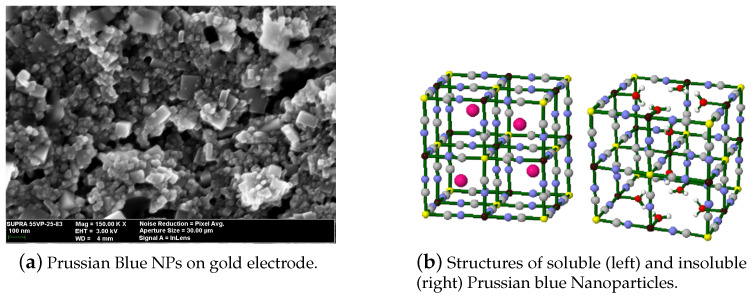
Characteristics of Prussian blue nanoparticles (PBNPs). (**a**) SEM micrograph of the PBNPs deposited by drop casting over an electrode; (**b**) Prussian blue crystalline structure. With permission from [18].

**Figure 3 sensors-21-02204-f003:**
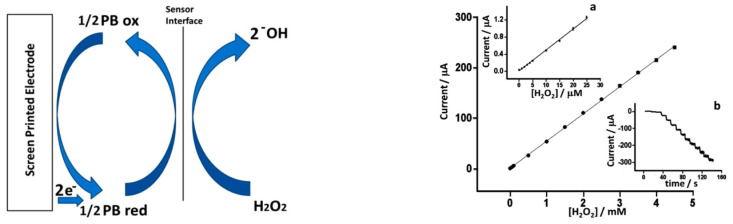
Electrochemical characterization of the electrodes with electrodeposited Prussian blue nanoparticles (PBNPs) layers. (**Left**) mechanism of the catalytic H${}_{2}$O${}_{2}$ reduction mediated by Prussian blue (PB), (**Right**) calibration curve of H${}_{2}$O${}_{2}$ on PBNP screen printed electrodes (SPEs). (**a**) Detail in the concentration range between 0 and 25 μM and (**b**) raw amperometric data from 0 to 4.5 mM. With permission from [29].

**Figure 4 sensors-21-02204-f004:**
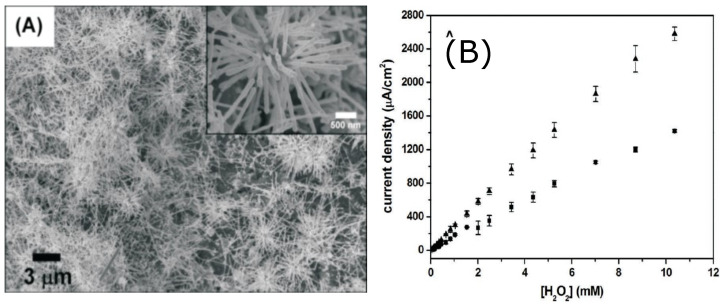
(**A**) SEM image of Ag urchin-like NWs on an SPC electrode (inset: enlarged view of a single cluster of urchin-like Ag NWs; (**B**) calibration curves of amperometric tests. Adapted with permission from [9].

**Figure 5 sensors-21-02204-f005:**
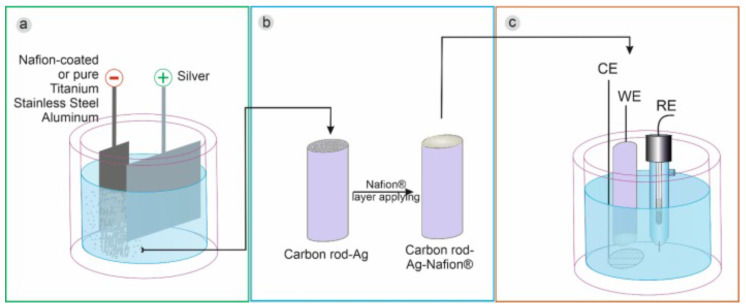
A general scheme for the synthesis of silver particles (**a**), their immobilization on the carbon rod electrode (**b**), and electrode utilization as a H${}_{2}$O${}_{2}$ sensor (**c**). With permission from [55].

**Figure 6 sensors-21-02204-f006:**
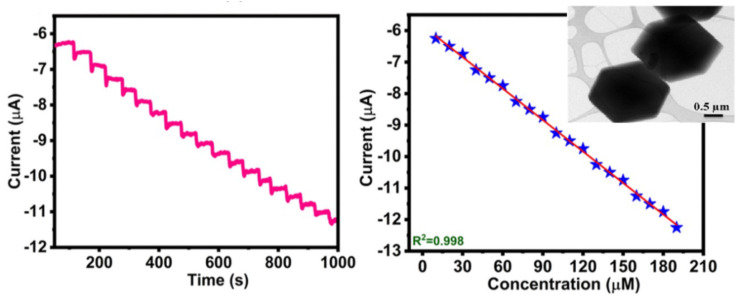
Chrono–amperometric response (**left**) of N-Co-MOF/GCE for the successive addition of 10 μM of H${}_{2}$O${}_{2}$ in 0.1 M PBS (pH = 7) under constant stirring, −0.35 V. To the (**right**), the figure shows the change in current at different concentrations of H${}_{2}$O${}_{2}$, the inset shows a TEM image of the Nitrogen-enriched MOF. Adapted with permission from [61].

**Figure 7 sensors-21-02204-f007:**
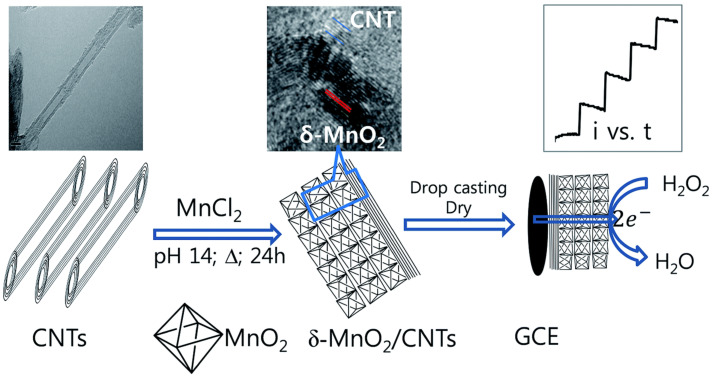
Scheme of the synthesis and application of a δ-MnO${}_{2}$ nanocomposite with carbon nanotubes (δ-MnO${}_{2}$/CNTs) as an enzyme-free sensor for the detection of hydrogen peroxide (H_2_O_2_) through an electroreduction reaction. (**Left**) carbon nanotubes; (**Middle**) synthesis of the δ-MnO${}_{2}$/CNTs nanocomposite by a simple one-step hydrothermal process in an alkaline solution without using surfactants or templates; (**Right**) investigation of the electrochemical properties of δ-MnO${}_{2}$/CNTs in a glassy carbon electrode by cyclic voltammetry and amperometry. With permission from [71].

**Figure 8 sensors-21-02204-f008:**
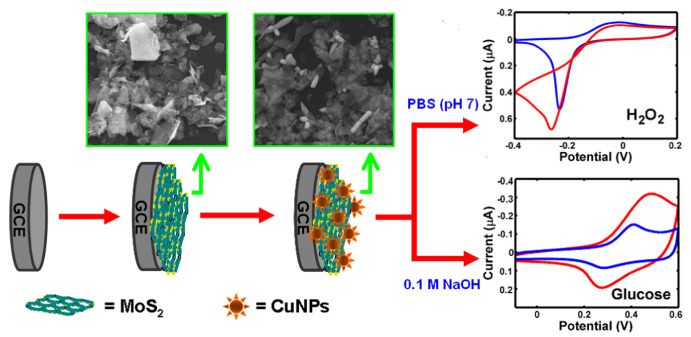
Modified glassy carbon electrode made of multilayer Cu/molybdenum disulfide (CuNFs-MoS${}_{2}$/GCE) with 3D nano-flower shape for non-enzymatic sensing. (**Left**) schematic representation of the fabrication of the CuNFs-MoS_2_/GCE, and (**Right**) possible reaction mechanisms at the electrode during the analysis for H${}_{2}$O${}_{2}$ and glucose. With permission from [81].

**Figure 9 sensors-21-02204-f009:**
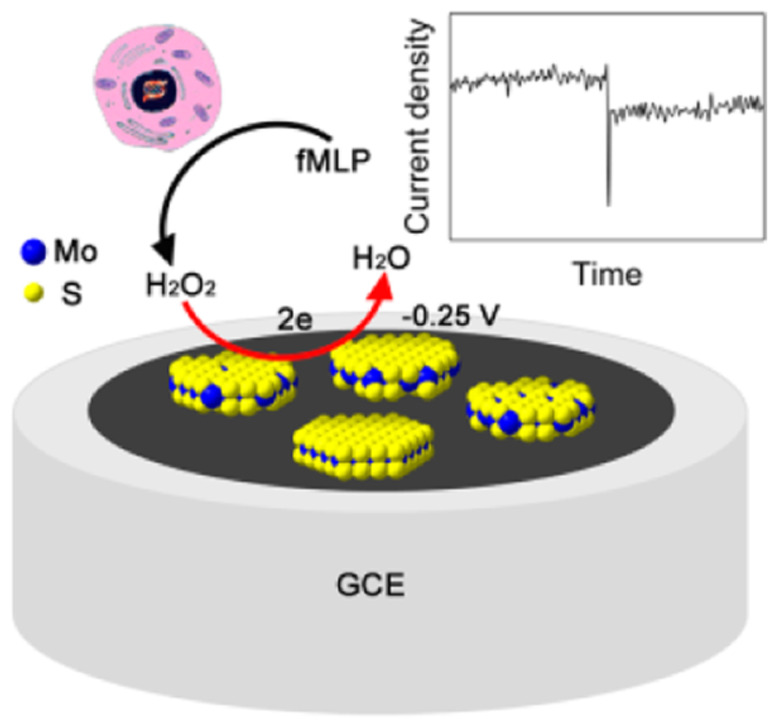
Schematic representation of a hydrogen peroxide H${}_{2}$O${}_{2}$ biosensor based on MoS${}_{2}$ nanoparticles. With permission from [82].

**Figure 10 sensors-21-02204-f010:**
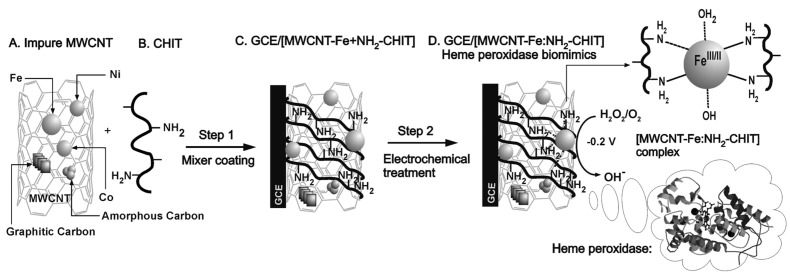
Schematic representation (**A**–**D**), of the construction of a chemically modified vitreous carbon electrode biosensor of impure multi-walled carbon nanotubes (MWCNT-Fe)-chitosan biopolymer (H${}_{2}$N-CHIT) (GCE/[MWCNT-Fe:H${}_{2}$N-CHIT]) for the electrocatalytic and electrochemical detection of hydrogen peroxide (H${}_{2}$O${}_{2}$) in phosphate buffer (PBS) pH 7. With permission from [95].

**Figure 11 sensors-21-02204-f011:**
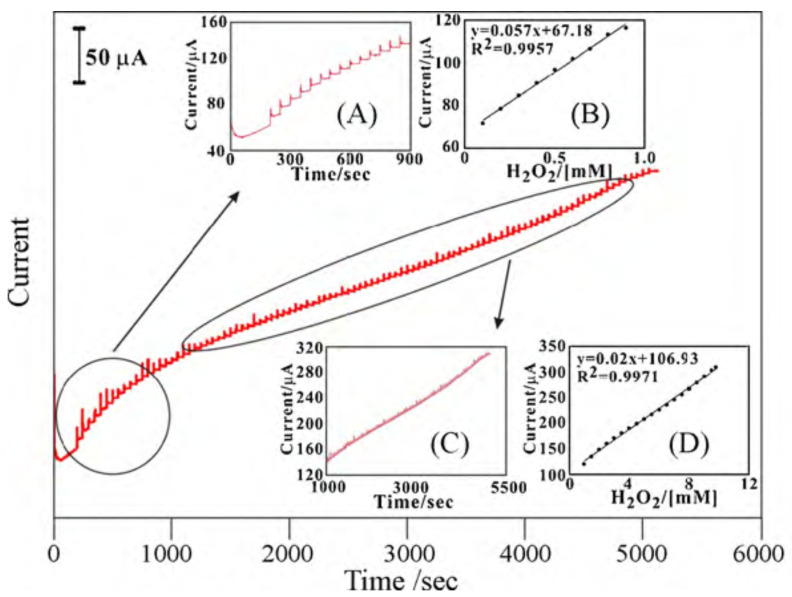
Amperometric responses of sequential additions of H${}_{2}$O${}_{2}$ at the MWCNT–PEDOT/GCE in pH 7 PBS. Rotating speed = 2000 rpm, E${}_{app.}$ = −0.5 V. The blank amperometric response of MWCNT–PEDOT/GCE is tested during 0–200 s. Inset: (**A**) scale-up amperometric response to the first H${}_{2}$O${}_{2}$ additions; (**B**) calibration plot of H${}_{2}$O${}_{2}$ concentration; (**C**) amperometric response to H${}_{2}$O${}_{2}$; (**D**) complete calibration plot. With permission from [96].

**Figure 12 sensors-21-02204-f012:**
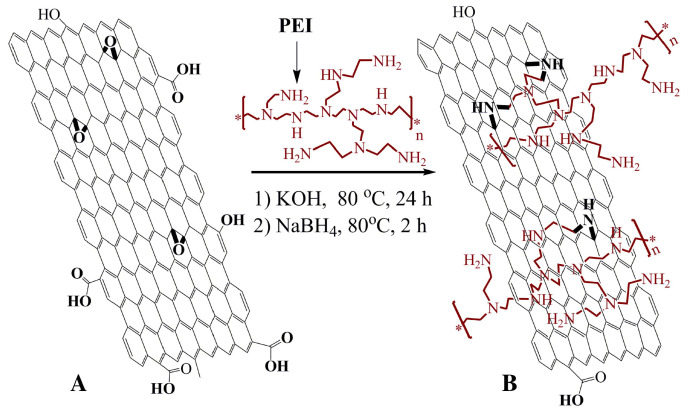
Schematic representation of the functionalization of graphene. (**A**) graphene oxide sheet; (**B**) PEI-funtionalized graphene. This composite was prepared firstly by stirring PEI and graphene in KOH at 80 ${}^{\circ }$C during 24 h. Then, NaBH${}_{4}$ was added to the mixture and kept at 80 ${}^{\circ }$C for 2 h. The PEI-functionalized graphene was collected by centrifugation and wash with distilled water. With permission from [97].

**Figure 13 sensors-21-02204-f013:**
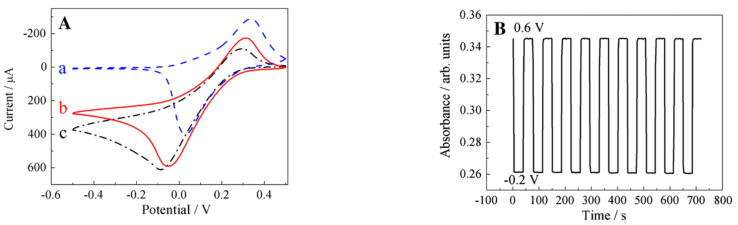
Electrochemical operation of the sensor built by the layer-by-layer complexation method. (**A**) Cyclic voltammograms of [PB/PEI–graphene]${}_{10}$ multilayer (**a**) in N${}_{2}$-saturated potassium hydrogen phthalate (0.1 M, pH = 4), [PB/PEI–graphene]${}_{10}$ multilayer; (**b**) and [PB/PEI]${}_{10}$ multilayer; (**c**) in the presence of 5 mM H${}_{2}$O${}_{2}$ in N${}_{2}$-saturated potassium hydrogen phthalate. Scan rate: 10 mV/s; (**B**) spectroscopic monitoring of potential switching for [PB/PEI–graphene]${}_{16}$ multilayer film immersed in a quiescent cuvette cell. Switching occurred between −0.2 V (Prussian white, low absorbance) and 0.6 V (Prussian blue, high absorbance). With permission from [97].

**Figure 14 sensors-21-02204-f014:**
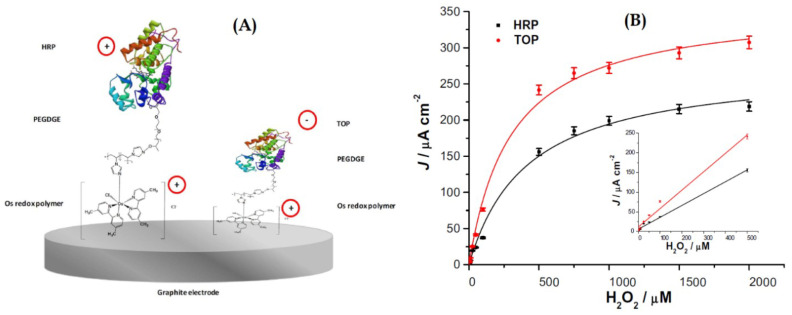
(**A**) Schematic representation of the biosensor based on HRP and the one based on TOP enzymes; (**B**) calibration curves for both biosensors in a FIA System with PBS buffer at pH 7 with 0.1 M KCl as carrier, and 20 μL sample injection volume, −0.1 V vs. Ag|AgCl (0.1 M KCl). The inset shows the linear section of the curves. With permission from [110].

**Figure 15 sensors-21-02204-f015:**
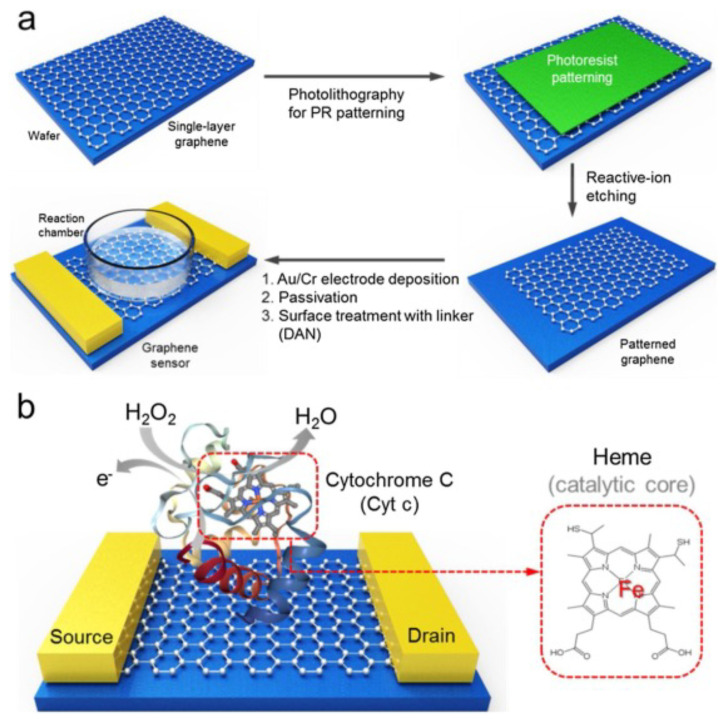
Schematic representation of the Cyt c modified single-layered graphene FET sensor for ultrasensitive H${}_{2}$O${}_{2}$ detection. (**a**) the overall fabrication process for graphene FET sensor platform. The fabrication strategy is first performed to obtain single-layered graphene-based FET platform for a high-performance sensor. The second step involves the conjugation of the Cyt c with surface-modified graphene via 1,5-diaminoaphthanlene (DAN). Then, this sensor platform was utilized for the detection of the H${}_{2}$O${}_{2}$ in real-time manner; (**b**) Cyt c-conjugated graphene FET sensor. Cyt c, which contains a heme structure as a catalytic site, is able to convert the value of the current via electron transport by H_2_O_2_-induced oxidation. With permission from [111].

**Table 1 sensors-21-02204-t001:** Metal hexacyanoferrates ${}^{b}$.

Nanomaterial	Transd. Princp.	Sensitivity [μA·mM${}^{-1}$·cm${}^{-2}$]	LR [mM]	LOD [μM]	Ref
PB@PtNPs/GF	Amp	40.9 × 10${}^{3}$	–	1.2 × 10${}^{-3}$	[25]
PB NPs	Amp	0.762 × 10${}^{3}$	0–4.5	0.2	[29]
PB-MWCNTs	Amp	0.436 × 10${}^{3}$	5–1645 × 10${}^{-3}$	0.35	[32]
PB	Amp	0.35 × 10${}^{3}$	up to 2.5	50	[39]
PB	CV	0.3 × 10${}^{3}$	–	0.5	[15]
GC-R/PB	Amp	0.25 × 10${}^{3}$	50 × 10${}^{-3}$–10	0.1	[21]
PB@Au NPs	Amp	39.72	2 × 10${}^{-3}$–8.56	0.1	[28]
PB-PPy NWs	Amp	10	0.2 × 10${}^{-3}$–7.2	–	[20]
NIHCF-GS	Photocurrent	3.53	2.0 × 10${}^{-3}$–2.3	1.0	[35]
CuHCF		2.34	up to 10	250	[39]
PB-PANI-HNTs	Amp	0.98	4–1064 × 10${}^{-3}$	0.226	[14]
ENM	Amp	0.237	up to 100 × 10${}^{-3}$	6.1	[40]
PB-RGO	Amp	0.1617	0.5 × 10${}^{-3}$–0.7	–	[27]
CPE/CFe*-RP	Amp	–	up to 0.8	33 × 10${}^{-3}$	[26]
CoHCNFPs-GR	Amp	–	0.6–379.5 × 10${}^{-3}$	0.1	[37]
PB	Amp	–	1 × 10${}^{-3}$–10	1	[16]
NiHCF	CV	–	0.2 × 10${}^{-3}$–1.5	1.2	[34]
CrHCF-SWNTs	Amp	–	0.5 × 10${}^{-3}$–10	–	[36]

^*b*^ Abbreviations: Amp: amperometry, CV: cyclicvoltametry.

**Table 2 sensors-21-02204-t002:** Metallic nanostructures based H${}_{2}$O${}_{2}$ sensors.

Nanomaterial	Transd. Princp.	Sensitivity [μA·mM${}^{-1}$·cm${}^{-2}$]	LR [mM]	LOD [μM]	Ref
Ag NWs	Amp	4.705 × 10${}^{3}$	50 × 10${}^{-3}$–10	10	[9]
nano Pd	Amp	1.42 × 10${}^{3}$	1–14 × 10${}^{-3}$	1	[50]
Ag NWs	Amp	749	0.2 to 1.5	46	[54]
PtRu	Amp	539.01	–	1.7	[53]
PtAu		415.46	–	2.0	[53]
PtIr		404.52	–	0.8	[53]
C${}_{rod}$@Ag-Ps	Amp	128	–	100	[55]
Au NPs	Amp	52.94	10 × 10${}^{-3}$–8	0.5	[59]
Pt NPs	Amp	9.15	0.5 × 10${}^{-6}$–4	500	[49]
TiN nanofilm	Amp	3.99	2 × 10${}^{-2}$–3	–	[57]
Pt NPs/SWCNT	Amp	3.57	25 × 10${}^{-6}$–10 × 10${}^{-3}$	25	[46]
Au/Pt NPs	Amp	2.92	10–80 × 10${}^{-3}$	10	[47]
Pt/TiO${}_{2}$	Amp	0.85	4 × 10${}^{-3}$–1.25	4	[48]
Ag NPs/ATP	Amp	–	10 × 10${}^{-3}$–21.53	2.4	[51]
Ag NPs/GO	Amp	–	0.1–20	1.9	[52]
Cu NPs/Chi/CNT	Amp	–	0.05–12	20	[42]
Cu NPs	Amp	–	8–70 × 10${}^{-3}$	3.4	[58]
TiN NRs	CV	–	0.5 × 10${}^{-3}$–2	0.5	[45]

**Table 3 sensors-21-02204-t003:** Sensors based on metal oxide nanostructures.

Nanomaterial	Transd. Princp.	Sensitivity μA·mM${}^{-1}$·cm${}^{-2}$	LR [mM]	LOD μM	Ref
NHGH/NiCo${}_{2}$O${}_{4}$	Amp	2072	1–510 × 10${}^{-3}$	0.136	[74]
Co${}_{3}$O${}_{4}$ NWs	Amp	1.14 × 10${}^{3}$	0.015–0.675	2.4	[72]
MnO${}_{x}$/CNW	Amp	698.6	40 × 10${}^{3}$–10	0.55	[83]
δ-MnO${}_{2}$/CNTs	Amp	243.9	0.05–22	1	[71]
CoOOH nanosheets	Amp	99	up to 1.6	40	[75]
Co doped ZnO NPs	Amp	92.4	–	14.3	[77]
ZrO${}_{2}$ NPs	Amp	82.13	0.05–0.25	–	[85]
PTBO/GCNT					
Cu${}_{2}$O	Amp	50.6	up to 1.5	1.5	[67]
MnO${}_{2}$	Amp	38.2	5–600 × 10${}^{-3}$	0.8	[69]
αFe${}_{2}$O${}_{3}$ NPs	Amp	21.62	1.0–44.0 × 10${}^{-3}$	0.4	[66]
CoMn${}_{2}$O${}_{4}$@GE	Amp	13.2	1–1000	40.2	[73]
TiO${}_{2}$/SiO${}_{2}$	Phosphorescence	–	7.0 × 10${}^{-3}$–70	–	[63]
CuO	Amp	–	5.0–180.0 × 10${}^{-3}$	1.6	[65]
α-Fe${}_{2}$O${}_{3}$ NRs	CV	–	40 × 10${}^{-3}$–4.66	–	[68]
MoS${}_{2}$ NPs	Amp	–	10${}^{5}$ × LOD	2.5 × 10${}^{-3}$	[82]
cobalt manganese oxide	Amp	–	0.1 to 25	15	[62]
MnO${}_{2}$	Amp	–	2.5 × 10${}^{-3}$–2.05	12	[70]
CuNFs/MoS${}_{2}$	Amp	–	0.04–1.88 × 10${}^{-3}$	0.021	[81]
CoO-CoS/NF	Amp	0.059	2–254 × 10${}^{-3}$	0.89	[76]

**Table 4 sensors-21-02204-t004:** Mixed nanostructures based H${}_{2}$O${}_{2}$ sensors.

Nanomaterial	Transd. Princp.	Sensitivity μA·mM${}^{-1}$·cm${}^{-2}$	LR [mM]	LOD μM	Ref
SG		202 × 10${}^{3}$	–	651.5	[98]
MWCNT–PEDOT	Amp	943	0.1–9.8	50	[96]
CVDG		173	–	15.1	[98]
rGO	Amp	25	–	9.2	[98]
PPy/PB NWs	Amp	10	0.2–7.2	–	[20]
HNTs	Amp	0.98	4–1064 × 10${}^{-3}$	0.226	[14]
MWCNT-Fe:H${}_{2}$N-CHIT	Amp	0.8345	50 to 2500 × 10${}^{-3}$	2.3	[95]

## Data Availability

Not applicable.

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
