# Peer review of "Nanostructures in Hydrogen Peroxide Sensing"

_sensors, 2021, doi:10.3390/s21062204_

Round 1
Reviewer 1 Report
In this work, the authors describe the developments in the field of electrochemical sensors based on nanomaterials for H2O2 detection. The use of various nanostructures based on noble metal nanoparticles, metal oxides and transition metals hexacyanoferrates, and carbon nanotubes is discussed with respect to the preparation methods and the analytical performance of the developed sensors. The combination of inorganic fillers with organic conducting polymers is also addressed. The comparison amongst various electrochemical sensors is based on some analytical performance figure of merit like limit of detection, linear response range and sensitivity. The use of enzymatic biosensors for H2O2 detection is also briefly described. The work is well organized and the discussion underlines the major advantages and disadvantages of each class of nanostructures used in H2O2 electrochemical sensing.
The manuscript can be considered for publication after minor revision:
- The literature survey includes only a limited number of references from the last 5 years. Despite the large number of cited references, the inclusion of some references from the last 5 years, selected according to the analytical performance criteria, will provide the reader with an updated overview on the electrochemical sensing of H2O2 using nanostructured materials.
- There should be consistency in the units of the linear ranges reported in various tables. For instance, the concentration ranges may be expressed in mol/L units.
Minor points:
Figure 2: the description in the legend should corroborate with the parts of this figure.
Page 4, line 96: please replace “polypirrol” by “polypyrrole”.
Author Response
Please, see the attachment.

Reviewer 2 Report
The review by Madrid et al., dealing with the hydrogen peroxide sensing through the nanosctructed materials, is rather well structured and organized and it can be published on this Journal after some minor issues will be amended, as I will say below.
The authors should go over the manuscript to check carefully both the typos and the English
In some parts there is a lack of sketches or Figures, as for example in the paragraph 2.2-Metallic nanostructures (from page 5 to 7) and in that 2.3 (page 8-9). In other parts I would enrich the review with some further few Figures of systems discussed in the text. This would improve the readability very much.
In the last part, section 2.5-Biomolecules, I would increase a bit the bibliography concerning the enzyme based devices with few more references published in recent years, as for example the work by Antiochia and Lo Gorton on Solid State Ionics 2018.
Author Response
Please, see the attachment.
